# Application of YOLO and ResNet in Heat Staking Process Inspection

**Hail Jung [1],\* and Jeongjin Rhee [2]**

1 Graduate School of Carbon Neutrality, Ulsan National Institute of Science and Technology (UNIST), Ulsan 44919, Republic of Korea
2 Data Science Group, INTERX, Seoul 08503, Republic of Korea
\* Correspondence: hail1995@unist.ac.kr; Tel.: +82-10-2994-7527

**Abstract:** In the automobile manufacturing industry, inspecting the quality of heat staking points in a door trim involves significant labor, leading to human errors and increased costs. Artificial intelligence has provided the industry some aid, and studies have explored using deep learning models for object detection and image classification. However, their application to the heat staking process has been limited. This study applied an object detection algorithm, the You Only Look Once (YOLO) framework, and a classification algorithm, residual network (ResNet), to a real heat staking process image dataset. The study leverages the advantages of YOLO models and ResNet to increase the overall efficiency and accuracy of detecting heat staking points from door trim images and classify whether the detected heat staking points are defected or not. The proposed model achieved high accuracy in both object detection (mAP of 95.1%) and classification (F1-score of 98%). These results show that the developed deep learning models can be applied to the real-time inspection of the heat staking process. The models can increase productivity and quality while decreasing human labor cost, ultimately improving a firm's competitiveness.

**Keywords:** object detection; classification; deep learning; Artificial Intelligence; heat staking process; manufacturing industry

## 1. Introduction

Technological advancement has enabled the development of various practical deep learning methodologies. Deep learning frameworks and architectures, such as YOLO (You Only Look Once) or ResNet, provide highly accurate and precise real-time identifications of objects [1]. These models have been used in solving on-site issues in diverse fields. This study attempts to further test the validity of recent deep learning models by identifying and classifying the quality of heat staking points. This study specifically focuses on the heat staking process of points on automobile door trims.

Employing deep learning-based quality prediction in the manufacturing process is particularly valuable because there are over sixty staking points in a single door trim and inspecting the quality of all points in a limited takt time is difficult. Furthermore, human errors are inevitable during measurements [2]. Because of these errors, acceptable product points are sometimes rejected (also known as "overkilled"), and defective product points are accepted as acceptable product points (also known as "escaped"). Both overkilling and escaping lead to tragic results, as overkill increases production costs and escape causes critical customer dissatisfaction.

One solution to partially alleviate these hurdles is to employ a machine vision system. The idea is to set up an environment for machine vision, take vision images, and run rule-based tests to check the quality of products. However, applying rule-based methods in a heat staking process is difficult because the locations and sizes of staking points vary from product to product. Therefore, it is recommended to employ a deep learning framework

that is relatively free from the problems of rule-based algorithms. In this manner, studies have employed various deep learning frameworks and discuss the results.

The problems of the current inspection process of heat staking points in the automotive industry are that the process fully relies on the human labor and that human labor often incorporates inspection errors. That is, due to various reasons such as immature work level, tiredness, and so on, inspection errors exist. Therefore, this study tried to employ various deep learning models to determine whether the artificial intelligence technology is an effective strategy to enhance the inspection process of heat staking points. In terms of the methodology, this paper tried to apply two different objectives—object detection and classification—and combine them into one deep learning model to apply in the inspection process. For the object detection, this study applied the YOLO methodology as it is a powerful algorithm and one frequently used in object detection problems. Using the YOLO network, this study detected all heat staking points, regardless of their quality. From the detected heat staking points, the study then used the ResNet classification model to further classify whether the detected heat staking point was defected or not. This process, all connected into one algorithm, can be a powerful alternative for heat staking manufacturing firms that have problems in inspection processes.

The advent of AlexNet was a huge turning point in deep learning applications [3]. Since the introduction of the AlexNet framework, models have been applied to various fields including the manufacturing industry [4,5]; however, slow detection rate was a practical problem when applying deep learning to the manufacturing industry. The YOLO series are representative one-stage detectors, and fast detection speed, a critical index in real-time application, is their most prominent feature [6–11].

Among the YOLO series, this study used the most recent, YOLOv5, for detecting staking points from automobile door trims and the ResNet classification model for classifying the quality of detected staking points. This study ensembled two different methods to increase both the efficiency and accuracy of the given task. In this study, the function of YOLOv5 was to localize the abnormal regions surrounding staking points. The advantage of using ResNet is in avoiding gradient explosion problems in deep learning for classification. Using both models, this study dynamically filtered the result from the YOLOv5 and ResNet models. Because of several advantages, prior studies have well-used object detection models and classification models simultaneously in solving specific given tasks [1,12–21].

Following prior findings, this study used YOLOv5 with ResNet and obtained robust results. Using 2400 training door trim images (100,310 staking points labeled) and 600 test images, this study first found that the training result of the YOLOv5x model was significantly accurate; the mAP was 0.951, precision was 0.934, and recall was 0.939. The highly accurate result showed the good ability of the YOLO model to detect the heat staking points in the door trim image. Furthermore, the ResNet classification model also showed a noteworthy result; the accuracy of the model was 0.98, and the F1-score (the harmonic mean of the precision and recall) was 0.98. These results imply that the ResNet model classified the quality of the staking points detected by the YOLOv5 model effectively. The high F1-scores also showed that the results were relatively free from type 1 or type 2 errors.

This paper highlights a technical innovation in the deep learning field. By joining the YOLO and ResNet models, it provides a novel method to more accurately simultaneously detect inspection points and classify their quality both accurately and reliably.

This research makes the following contributions: first, it applies a deep learning framework to a real-time problem, particularly in the heat staking process. Inspecting the quality of a manufactured product and guaranteeing high quality for customers are critical for a business's sustainable growth. Manual inspection has long encompassed problems. Immature work skills because of frequent labor changes and increased process complexity are typical examples that lead to human errors in the inspection process [4]. With the necessity to employ a deep learning-based vision system into the manufacturing process, this paper shows that the combination of the YOLO and ResNet frameworks

can reduce costs and ultimately increase productivity. This paper also contributes to the literature on applying ensemble methods, particularly by combining objection detection and classification methodologies. It shows that the model's performance is improved to the extent that it could be used on a real-time heat staking process.

The rest of this paper is organized as follows: In Section 2, it reviews relevant studies to this system and discusses key takeaway messages. It outlines the object detection and classification framework in Section 3. It provides the experimental results, consisting of object detection and quality classification on a real dataset, in Section 4. Finally, Section 5 discusses the results and provides potential recommendations for future research.

## 2. Related Work

### 2.1. YOLO Framework

The YOLO framework, a popular object detection model, was introduced in 2016. YOLO refers to the ability of the human visual system to immediately detect objects. Therefore, the YOLO framework was designed to detect objects similarly to the human visual system. The first YOLO consisted of a 24-layer convolutional neural network for feature extraction and two fully connected layers for predicting the probability and coordinates of objects. The latest version, YOLOv5, outperforms previous YOLO versions.

Because of its intuitiveness and innovativeness, the framework is renowned for its fast speed and accuracy. The YOLO framework introduces a new structure for object recognition systems and has received much attention; accordingly, it has been widely used and applied in various applications [22–43].

### 2.2. ResNet Framework

The ResNet framework is a residual neural network, which is a gateless or open-gated variant of the HighwayNet, a deep feedforward neural network with hundreds of layers [44]. The ResNet classification models are implemented with double layer skips that contain nonlinearities (ReLU) and batch normalization in between. Like the long short-term memory (LSTM) model, ResNet skips connections to avoid the gradient vanishing problem (leading to easier optimization of neural networks) or to mitigate the degradation problem [45]. Accuracy saturates when additional layers in a neural network increase training errors [44].

This skipping effectively simplifies and lightens the neural network, and the neural network learns by reducing the impact of vanishing gradients, as there are fewer layers to propagate through. The network then gradually restores the skipped layers. When all layers are expanded, it stays closer to the manifold. A neural network without residual parts explores more of the feature space. This is more vulnerable to perturbations, and it thus necessitates additional training to recover. Because of its advantages, ResNet is widely used and is one of the most cited neural network frameworks. Specifically, it is widely accepted in industries where takt time and accuracy are two important criteria. As ResNet is lighter than other deep learning-based classification models but a powerful classification learner, researchers and practitioners have long developed and employed it in their domain [10,11,14,15,46–52].

## 3. Proposed System

### 3.1. Shortcomings of Existing Models

There are small, multiple points in each door trim that require heat staking. In fact, in this sample dataset, although the number of points varied among automobile types, each image had approximately over sixty points. Multiple object detection and recognition tasks are important topics in manufacturing. Figure 1a shows the outline of the model for applying YOLO to heat staking process inspection. If this study employed only the YOLO model, then for each input door trim image it would search for points and show bounding boxes with a confidence level. Final detection results would be shown accordingly. However, YOLO frameworks have shortcomings in identifying positions of multiple objects

accurately. Furthermore, the accuracy of classifying multiple small objects is low, which also reduces the recall rate, ultimately causing problems with adopting YOLO models on real heat staking process sites. Figure 1b illustrates the outline of the classification model ResNet. If this study only employed the ResNet model to the problem, then it would be able to classify whether the door trim image was of a defect or not, but the classification model has difficulties in illustrating where the defect parts are. Thus, for industrial usage where the workers must understand where the defects are located, classification models are hard to use.

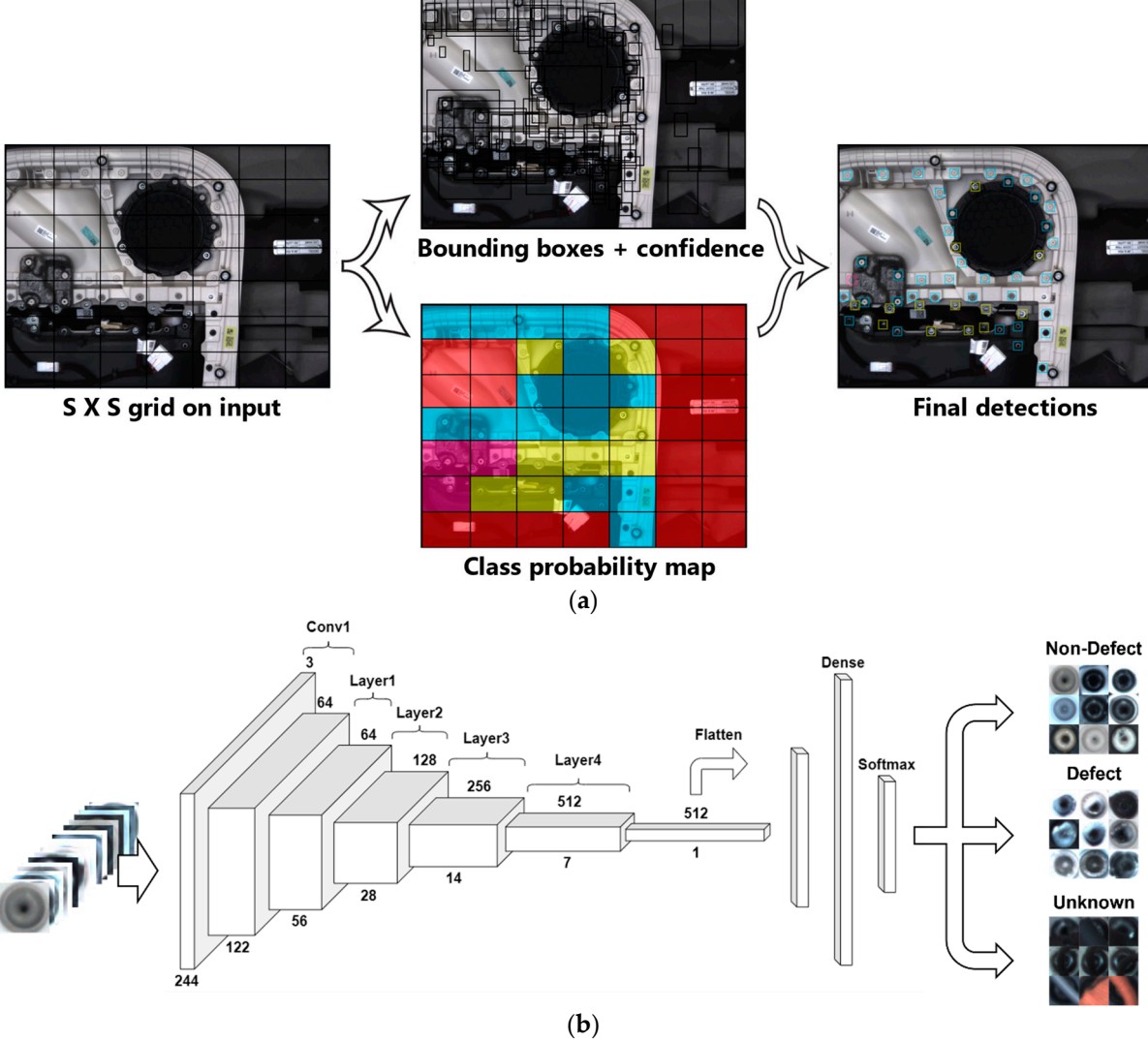

**Figure 1.** (**a**) YOLO flowchart for detecting quality of heat staking points in door trim images and (**b**) ResNet flowchart for classifying the quality of heat staking points.

To mitigate the issue to an extent, this study used the improved and modified YOLOv5 model by adding a deep ResNet framework with the same number of layers as the Darknet network in the feature extraction part of YOLOv5 [53]. Afterwards, the mean value was reduced to generate feature graphs of three scales after outputting the two feature extraction models. This process enabled more efficient information extraction of heat staking points in the door trim images and was more efficient in conducting object detection. Figure 2a shows an example of a full door trim image, and Figure 2b illustrates the real door trim images with heat staking points boxed around. Note that the heat staking points take a small portion of the entire door trim image.

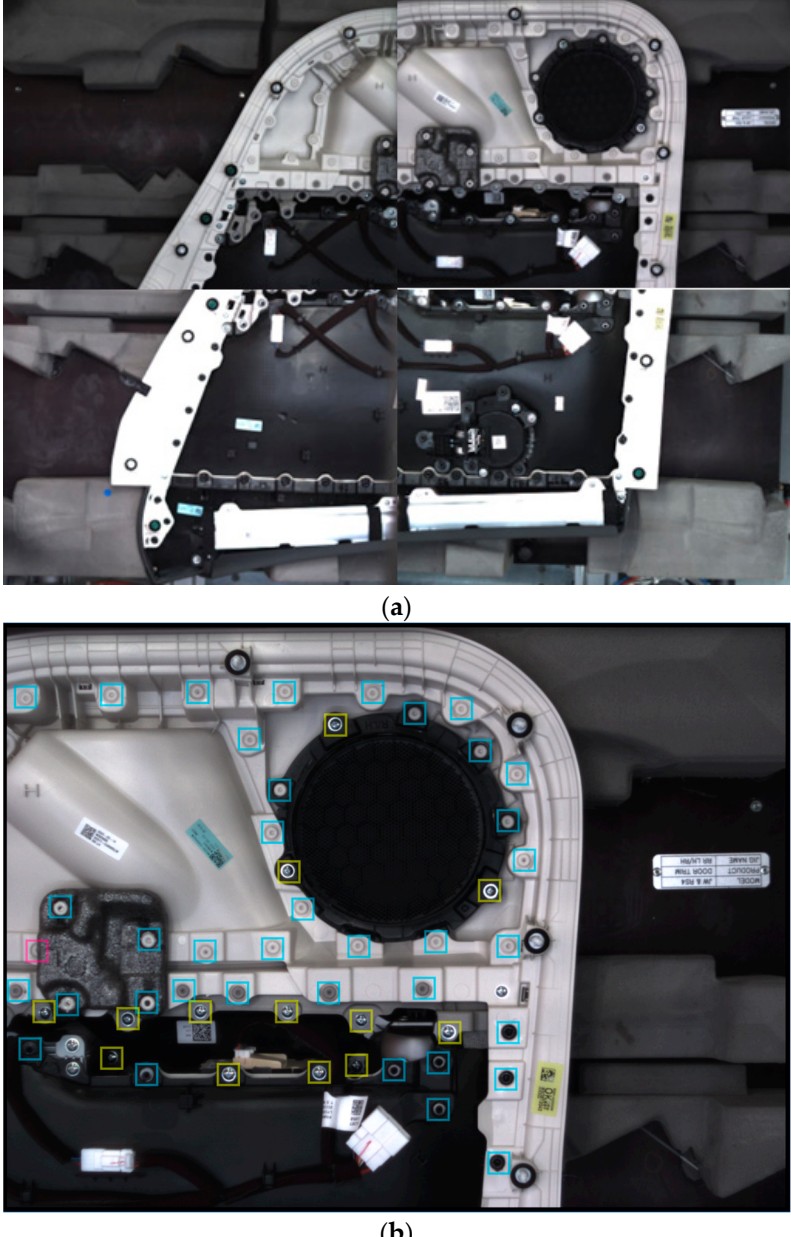

**Figure 2.** Example image of a (**a**) full door trim and (**b**) detected heat staking points.

Figure 3 illustrates the larger size of the heat staking points grouped by quality. For the classification, this study manually grouped the images into three different categories. The images were collected with a camera with a resolution of 5 M and an effective number of pixels (H × V) of 2592 × 1944: Figure 3a shows good quality points, Figure 3b shows defected points, and Figure 3c shows unknown points. Unknown points consist of points that are either partially covered by other cables, or out-of-focus and blurred images. Because the images are small-sized and the differences between defects and non-defects are not large, deepening the neural network will lead to the gradient vanishing problem. Therefore, this study chose ResNet, as it has the advantage of solving the gradient vanishing problem while deepening the network. This is typically important in the inspection of the heat staking process because the heat staking points are extremely small compared with the door trim images.

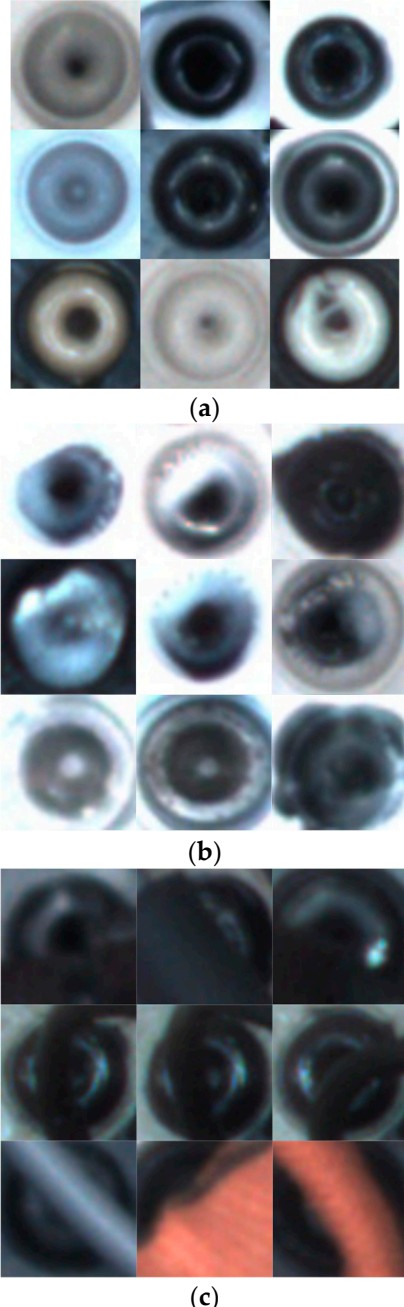

**Figure 3.** Example of a (**a**) non-defected heat staking points, (**b**) defected heat staking points, and (**c**) unknown points.

### 3.2. Suggested YOLO-ResNet Model

Figure 4 shows the flowchart of the proposed network that combines YOLOv5 and ResNet. Based on the Darknet network structure for feature extraction, ResNet was added for feature extraction, which solved the problem of poor accuracy in object detection.

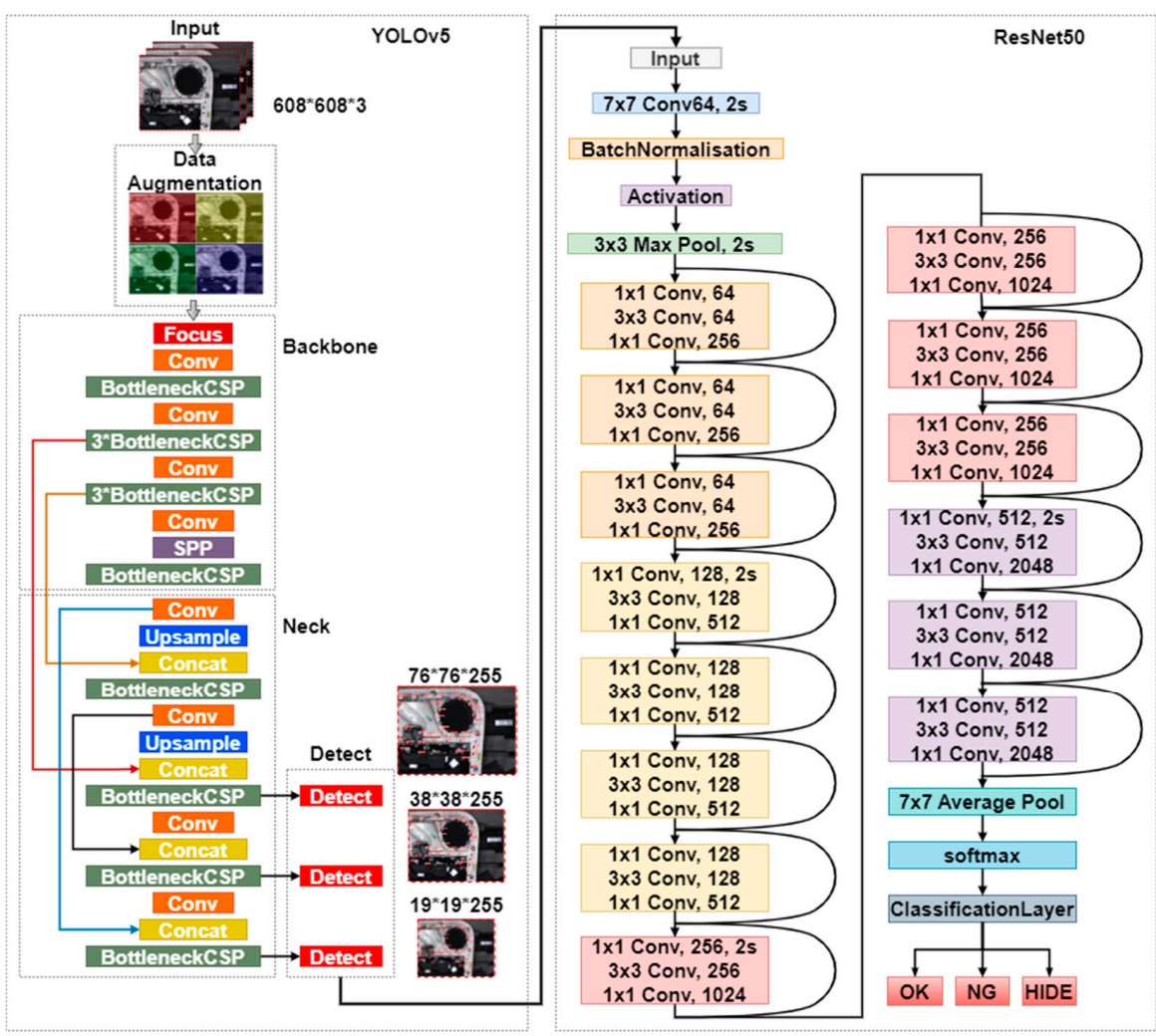

**Figure 4.** Flowchart of the proposed model, YOLOv5 with ResNet.

The model is composed of six main parts: a deep fully convolutional network, region proposal network, ROI pooling and fully connected networks, bounding box regressor, classifier for object detection and ResNet classifier. For consistency, this paper employed a deep fully convolutional network. The input image was put into the initial stage and extracted to 256 × n × n feature map, which is the input of region proposal network and ROI pooling layer. In the region proposal network, there are k anchors with different scales and ratios for each point on feature map. There will be n × n × k candidate windows that are ranked according to the score, and then 2000 candidate windows are obtained through non-maximum suppression. Overall, complexity is O(N^2/2), which is consistent with the classic YOLO model. This is because the ResNet classifier was attached as a final layer to the current YOLO network.

Furthermore, in terms of the industrial usage, this proposed network is useful. That is, using solely the YOLO model results in difficulties in classifying points by their quality, and using only the image classification model results in difficulties in visualizing where the defect points are located. In this manner, taking only the advantages of both the object detection and the classification models enables to first detect the heat staking points, and then use the classifier to classify whether the detected points are defected or not.

## 4. Experimental Results

### 4.1. Experimental Dataset

This study used a novel and rich dataset provided by SEOYON E-HWA, from South Korea. The company's main products are interior parts such as door trims, consoles, head linings, and package trays inside automobiles. They also manufacture exterior parts such as bumpers and seats for commercial vehicles. Thanks to their cooperation, this study was able to retrieve 3000 door trim images that had over sixty heat staking points. This study split the collected images such that there were 2400 training images and 600 test images.

### 4.2. Experimental Results

Table 1 first provides the hyperparameter tuning results for five different schemes (One to Five). Following prior YOLO literature, this study tuned the following hyperparameters that have been frequently tuned in past studies, to the five different schemes: lr0, lrf, momentum, weight_decay, warmup_epochs, warmup_momentum, warmup_bias_lr, box, cls, cls_pw, obj, obj_pw, iou_t, anchor_t, hsv_h, hsv_s, hsv_v, translate, scale, fliplr, and mosaic. The hyperparameter settings in the fourth scheme provided the best performance. Table 1 also shows the entire hyperparameter values and their descriptions. For easier crosschecking with other related literature, this study followed the default hyperparameter value settings. The trained model showed a precision of 0.995, a recall of 0.996, and a mAP@.5 of 0.994. The results imply that the parameters are well tuned.

**Table 1.** Hyperparameter tuning results of YOLO model.

| Scheme | Optimizer | Class | Images | Labels | Precision | Recall | mAP@.5 | mAP@ |
|---|---|---|---|---|---|---|---|---|
| One | AdamW | All | 600 | 25,999 | 0.983 | 0.989 | 0.989 | 0.652 |
| | | Points | | 16,163 | 0.989 | 0.999 | 0.99 | 0.699 |
| | | Screw | | 9523 | 0.98 | 0.996 | 0.992 | 0.679 |
| | | Hide | | 313 | 0.981 | 0.971 | 0.986 | 0.578 |
| Two | AdamW | All | 600 | 25,999 | 0.946 | 0.963 | 0.961 | 0.596 |
| | | Points | | 16,163 | 0.98 | 0.998 | 0.988 | 0.658 |
| | | Screw | | 9523 | 0.972 | 0.993 | 0.991 | 0.646 |
| | | Hide | | 313 | 0.886 | 0.898 | 0.903 | 0.485 |
| Three | AdamW | All | 600 | 25,999 | 0.986 | 0.969 | 0.977 | 0.594 |
| | | Points | | 16,163 | 0.99 | 0.997 | 0.988 | 0.645 |
| | | Screw | | 9523 | 0.985 | 0.987 | 0.99 | 0.624 |
| | | Hide | | 313 | 0.983 | 0.923 | 0.953 | 0.514 |
| Four | AdamW | All | 600 | 25,999 | 0.995 | 0.996 | 0.994 | 0.664 |
| | | Points | | 16,163 | 0.998 | 0.999 | 0.995 | 0.693 |
| | | Screw | | 9523 | 0.995 | 0.996 | 0.995 | 0.647 |
| | | Hide | | 313 | 0.993 | 0.994 | 0.994 | 0.651 |
| Five | AdamW | All | 600 | 25,999 | 0.992 | 0.985 | 0.99 | 0.744 |
| | | Points | | 16,163 | 0.997 | 1 | 0.995 | 0.785 |
| | | Screw | | 9523 | 0.996 | 0.994 | 0.995 | 0.782 |
| | | Hide | | 313 | 0.984 | 0.962 | 0.98 | 0.665 |

| Hyperparameters | Description | | Value |
|---|---|---|---|
| lr0 | Initial learning rate | | 0.01 |
| lrf | Final OneCycleLR learning rate (lr0 * lrf) | | 0.01 |
| momentum | Tuning parameter for the gradient descent algorithm | | 0.937 |
| weight_decay | Optimizer weight decay | | 0.0005 |
| warmup_epochs | Warmup epochs | | 3.0 |
| warmup_momentum | Warmup initial momentum | | 0.8 |
| warmup_bias_lr | Warmup initial bias learning rate | | 0.1 |
| box | Box loss gain | | 0.05 |
| cls | Class loss gain | | 0.5 |

**Table 1.** *Cont.*

| Scheme | Optimizer | Class | Images | Labels | Precision | Recall | mAP@.5 | mAP@ |
|--------|-----------|-------|--------|--------|-----------|--------|--------|------|
| cls_pw | | | Class BCELoss positive weight | | | | 1.0 | |
| obj | | | Object loss gain (scale with pixels) | | | | 1.0 | |
| obj_pw | | | Object BCELoss positive weight | | | | 1.0 | |
| iou_t | | | IoU training threshold | | | | 0.20 | |
| anchor_t | | | Anchor-multiple threshold | | | | 4.0 | |
| hsv_h | | | Image HSV-Hue augmentation (fraction) | | | | 0.015 | |
| hsv_s | | | Image HSV-Saturation augmentation (fraction) | | | | 0.7 | |
| hsv_v | | | Image HSV-Value augmentation (fraction) | | | | 0.4 | |
| translate | | | Image translation (+/− fraction) | | | | 0.1 | |
| scale | | | Image scale (+/− gain) | | | | 0.5 | |
| fliplr | | | Image flip left-right (probability) | | | | 0.5 | |
| mosaic | | | Image mosaic (probability) | | | | 1.0 | |

This study trained the YOLO model on the training dataset that included 2400 door trim images, which consisted of 200,620 heat staking points. The accuracy of the trained model was then evaluated on the test set of 600 images. Table 2 presents the results that show that YOLOv5m outperformed YOLOv5n, YOLOv5s, YOLOv5l, and YOLOv5x, with a precision of 0.947, F1-score of 0.930, and mAP@.5 of 0.956. These results are significant. It should be noted that the performance of the YOLO v5m model outperformed the YOLO v5l and YOLO v5x model. Theoretically, YOLO v5l and YOLO v5x should outperform the YOLO v5m model as v5l and v5x are larger size in terms of the number of extracted features thus are able to train deeper. However, the results show that the optimum number of features to be extracted do not monotonically increase. This implies that there are a smaller number of features to be extracted from images of heat staking points, and that a certain number of features provided by YOLO v5m model is enough. Therefore, this study used the optimal model, YOLO v5m, as the baseline object detection model which was then used to merge with the classification model.

**Table 2.** YOLO model heat staking points detection results.

| Model | Class | Opt. | Images | Labels | Precision | Recall | mAP@.5 | mAP@ |
|-------|-------|------|--------|--------|-----------|--------|--------|------|
| Yolov5n | All | AdamW | 600 | 100,310 | 0.937 | 0.914 | 0.945 | 0.592 |
| | Points | | | 54,060 | 0.978 | 0.99 | 0.992 | 0.622 |
| | Screw | | | 43,500 | 0.972 | 0.989 | 0.986 | 0.692 |
| | Hide | | | 2750 | 0.863 | 0.762 | 0.857 | 0.461 |
| Yolov5s | All | AdamW | 600 | 100,310 | 0.924 | 0.944 | 0.945 | 0.592 |
| | Points | | | 54,060 | 0.978 | 0.99 | 0.992 | 0.622 |
| | Screw | | | 43,500 | 0.969 | 0.993 | 0.987 | 0.692 |
| | Hide | | | 2750 | 0.826 | 0.848 | 0.855 | 0.461 |
| Yolov5m | All | AdamW | 600 | 100,310 | 0.947 | 0.93 | 0.956 | 0.591 |
| | Points | | | 54,060 | 0.987 | 0.987 | 0.992 | 0.627 |
| | Screw | | | 43,500 | 0.98 | 0.988 | 0.987 | 0.689 |
| | Hide | | | 2750 | 0.874 | 0.816 | 0.89 | 0.456 |
| Yolov5l | All | AdamW | 600 | 100,310 | 0.934 | 0.929 | 0.952 | 0.594 |
| | Points | | | 54,060 | 0.982 | 0.99 | 0.992 | 0.614 |
| | Screw | | | 43,500 | 0.979 | 0.989 | 0.988 | 0.689 |
| | Hide | | | 2750 | 0.841 | 0.808 | 0.877 | 0.479 |
| Yolov5x | All | AdamW | 600 | 100,310 | 0.934 | 0.939 | 0.951 | 0.595 |
| | Points | | | 54,060 | 0.984 | 0.988 | 0.99 | 0.619 |
| | Screw | | | 43,500 | 0.982 | 0.99 | 0.988 | 0.696 |
| | Hide | | | 2750 | 0.836 | 0.839 | 0.876 | 0.469 |

As the object detection results are significant, this study used the obtained heat staking points to further develop the ResNet classification model. Table 3 and Figure 5 report the summary statistics of the ResNet classification model. It should be noted that the F1-scores for non-defected, defected, and unknown classes were generally above 0.97. In the manufacturing industry, it is important to obtain a high F1-score, as recall and precision are both important factors [4]. This classification model's high accuracy shows that ResNet well captured the different distribution of features between defected and non-defected heat staking points of automobile door trims. Furthermore, Table 4 reports performance metrics of the YOLO-ResNet model with the statistics for recall, false negative rate, precision, false negative rate, and F1-score of the model. It should be noted that the overall performance metrics is generally high and significant. Specifically, important metrics for manufacturing firms such as recall, and precision are approximately 98%. The overall results show that the generated YOLO-ResNet model is a significant model that may somewhat replace human labor inspection process.

**Table 3.** ResNet model heat staking points quality classification results.

|            | Precision | Recall | F1-Score | Support |
|------------|-----------|--------|----------|---------|
| Non-defect | 0.97      | 0.99   | 0.98     | 306     |
| Defect     | 0.96      | 0.93   | 0.95     | 105     |
| Unknown    | 0.99      | 0.97   | 0.98     | 160     |
| Accuracy   |           |        | 0.98     | 571     |

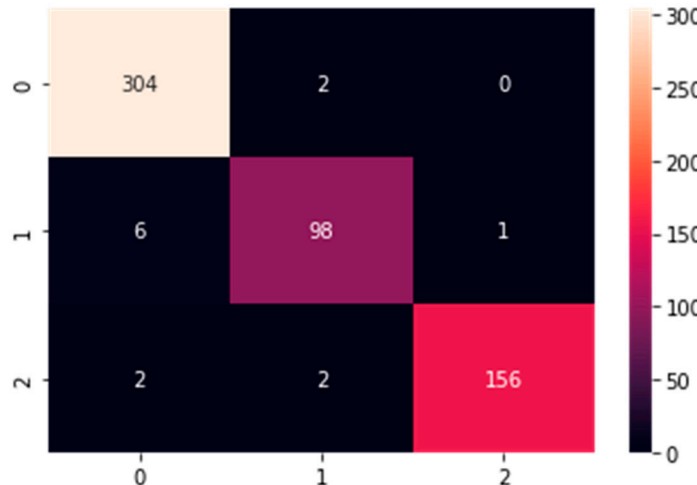

**Figure 5.** Confusion matrix of the ResNet classification model.

**Table 4.** YOLO-ResNet model performance metrics.

$$Recall\ (R) = \frac{TP}{TP+FN} = 0.98$$
$$False\ Negative\ Rate\ (FNR) = 1.00 - R = 0.02$$
$$Precision\ (P) = \frac{TP}{TP+FP} = 0.98$$
$$False\ Negative\ Rate\ (FPR) = 1.00 - P = 0.02$$
$$F1\ score = \frac{2 \cdot P \cdot R}{P+R} = 0.98$$

## 5. Conclusions

Increased labor cost adds to the overall costs suffered by manufacturing firms in developed nations. One potential breakthrough to overcome the issue is to employ Artificial Intelligence models in their manufacturing process. Many studies have investigated the usage of object detection and image classification in deep learning models. However, there are few studies on the application of these models in the inspection of the heat staking process. In this respect, this study proposed a new YOLO–ResNet model for detection

and classification of heat staking points in the automobile manufacturing industry. In this model, the YOLOv5 framework first detects the heat staking point accurately and then ResNet classifies whether the detected points are defected or not. The ability to detect the points and classify their quality is sufficiently accurate to be applied to real manufacturing sites.

This study used YOLO and ResNet models to detect the heat staking points and classify their quality. In the future, it is recommended to apply other object detection models or classification models and improve the current models. Furthermore, future researchers may also consider adding more defect categories for the heat staking process. The current model only considered whether the detected images are defected, non-defected, or unknown. However, there are multiple defect categories in the heat staking process such as overstaking and understaking. More accurate categorization may be beneficial for industries.

This study makes a contribution to the manufacturing industry that suffers from high inspection costs. The F1-score of the model is well-above 97%, which implies that if the model is used in the manufacturing inspection process, it would perform like or better than the human labor inspection system. Furthermore, if the model can train multiple defect types of heat staking points, then classifying detected heat staking points into different categories by their defect type would also be possible. Such multi-class object detection and classification hybrid model would be an important asset for the sustainable development and growth of manufacturing firms.

Furthermore, this study makes a contribution to the growing deep learning application literature. There are recent studies that employ image classification, multi-strategy particle swarm and ant colony hybrid optimization and optimal search mapping on various fields [54–57]. The underlying mechanism is to use modern algorithms and understand how to employ them in a certain domain field. In this manner, this study contributes by showing how YOLO and ResNet model could be used in the manufacturing inspection process.

**Author Contributions:** Conceptualization, H.J.; methodology, H.J.; software, J.R.; validation, J.R.; formal analysis, J.R.; investigation, J.R.; resources, H.J.; data curation, H.J.; writing—original draft preparation, H.J.; writing—review and editing, H.J.; visualization, H.J.; supervision, H.J.; project administration, H.J.; funding acquisition, H.J. All authors have read and agreed to the published version of the manuscript.

**Funding:** This research was funded by Korea Technology and Information Promotion Agency for SMEs (TIPA) funded by the Korean Government [Ministry of SMEs and Startups (MSS)] under Grant RS-2022-00140573.

**Institutional Review Board Statement:** Not applicable.

**Informed Consent Statement:** Not applicable.

**Data Availability Statement:** The heat staking points dataset is available upon request. Please contact the corresponding author for data.

**Conflicts of Interest:** The authors declare no conflict of interest.

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
