# Peer review of "Application of YOLO and ResNet in Heat Staking Process Inspection"

_sustainability, doi:10.3390/su142315892_

Round 1
Reviewer 1 Report
In principle, I agree with the publication of this scientific paper.
In my opinion, this research is still far from its use in the automotive industry, because all possible defects that occur in the manufacturing process using heat staking points technology must be evaluated simultaneously. The solution proposed by the authors of the paper cannot detect defects such as overstaking and understaking heat staking points. This is also revealed by the authors in the conclusions presented in the paper.
However, the work represents a step forward through the YOLO-ResNet model.
Author Response
Comment: In principle, I agree with the publication of this scientific paper. In my opinion, this research is still far from its use in the automotive industry, because all possible defects that occur in the manufacturing process using heat staking points technology must be evaluated simultaneously. The solution proposed by the authors of the paper cannot detect defects such as overstaking and understaking heat staking points. This is also revealed by the authors in the conclusions presented in the paper. However, the work represents a step forward through the YOLO-ResNet model.
Response: Thank you for your sincere concern and comment. We agree with your concern that the proposed solution yet has difficulties in detecting defects such as overstaking and understaking heat staking points. We have tries to further segment the defects by its type, however the accuracy of the developed model was not significant enough to be used in the automotive industry. Thus, we have decided to widen the research scope and focus whether the proposed YOLO-ResNet deep learning model may at least differentiate defects and non-defects. At the same time, we are currently also further researching technologies to successfully distinguish overstaking and understaking points. Once again, thank you for the comment.
Reviewer 2 Report
The manuscript “Application of YOLO and ResNet in Heat Staking Process In-Spection”, applied an object detection algorithm, the You Only Look Once (YOLO) framework, and a classification algorithm, residual network (ResNet), on a real image heat staking process image dataset.
In general I foud the manuscript difficult to understand, a) What is the problem statement? b) the methodology used, the details of the measurements and the application of the method are no clear, c) There is no discussion of the experimental results, only three paragraphs!!.
Others observations
1. In scientific articles, first person plural pronouns are not used.
2. Figure 1 is used to in identify positions of multiple objects accurately, but what about to detecting quality of heat staking points?
3. Figure 3, what kind of camara was used?, there are a few discussion.
4. Fig 4 too small and few discussion
5. Table 1, caption is not clear, the hyperparameters are no defined, there are more than five!!
6. Table 2, there are no discussion.
Author Response
We greatly appreciate the review of our paper and are grateful for the referees’ insightful and constructive comments. We have addressed all the comments and below, we outline our point-by-point response to each comment. After the revisions, the paper has improved significantly.
Responses to reviewer 2
Comment 1: In general, I found the manuscript difficult to understand, a) What is the problem statement? b) the methodology used, the details of the measurements and the application of the method are no clear, c) There is no discussion of the experimental results, only three paragraphs!
Response: Thank you for the comment. We agree that the previously submitted manuscript had difficulties for readers to understand. Following your suggestion, we have concretely described the problem statement, methodology used and the details of the application.
The problems of the current inspection process of heat staking points in the automotive industry are that the process fully relies on the human labor and that human labor of-ten incorporates inspection errors. That is, due to various reasons such as immature work level, tiredness, and so on, the inspection errors exist. Therefore, this study tries to employ various deep learning models to see if the artificial intelligence technology is an effective strategy to enhance the inspection process of heat staking points. In terms of the methodology, this paper tries to use two different architecture - object detection and classification – and combine them into one deep learning model to apply in the inspection process. For the object detection, this study applies YOLO methodology as it is a powerful algorithm and frequently used one in the object detection problems. Using YOLO network, this study detects all heat staking points, regardless of their quality. From the detected heat staking points, the study then uses ResNet classification model to further classify whether the detected heat staking point is defected or not. This process, all connected to one algorithm, can be a powerful alternative for heat staking manufacturing firms that have problems in inspection processes.
This study also has a strong contribution to the manufacturing industry that suffers from high inspection costs. F1-score of the model is well-above 97%. The result implies that if the model is used in the manufacturing inspection process, it would perform like or better than human labor inspection system. Furthermore, if the model can train multiple defect types of heat staking points, then classifying detected heat staking points into different categories by their defect type would also be possible. Such multi-class object detection and classification hybrid model would be an important asset for manufacturing firm’s sustainable development and growth.
We deeply appreciate your comment. We have largely edited the abovementioned problem statements, methodologies, and discussions on experimental results. Please refer to the lines 48 to 62 for problem statements and methodologies, lines 319 to 326 for discussions on experimental results.
Comment 2: In scientific articles, first person plural pronouns are not used.
Response: Thank you for the comment. We agree that scientific articles do not use first person plural pronouns. Following your suggestion, we have removed first person pronouns and replaced them with relevant terminologies.
Comment 3: Figure 1 is used to in identify positions of multiple objects accurately, but what about to detecting quality of heat staking points?
Response: We appreciate your concern that Figure 1 only illustrated the YOLO architecture but not classification model (ResNet). Following your suggestion, we have drawn how the previous quality inspection process would have been when only classification model is used. Figure 1b illustrates the outline of the classification model ResNet. If we only employ the ResNet model into problem, then we would be able to classify whether the door trim image is a defected image or not, but the classification model has difficulties in illustrating where the defect parts are. Thus, for industrial usage where the workers must understand where the defects are located, classification models are hard to be used. The illustrated image is as follows (Figure 1b), and please refer to the lines 162 to 167 for modifications.
Comment 4: Figure 3, what kind of camara was used? there are a few discussions.
Response: Thank you for the comment. The images were collected with a camera with a resolution of 5M and an effective number of pixels (H×V) of 2,592×1,944. We also have included the above camera specifications into the article. Please refer to lines 195 to 197 for changes.
Comment 5: Fig 4 too small and few discussion
Response: Following your suggestion, we have enlarged the Figure 4 to increase the visibility. We have also included the updated figure into the response letter. Furthermore, we added discussions regarding the Figure 4. Figure 4 shows the proposed network that combines YOLOv5 and ResNet. The flowchart illustrates the process of combining the YOLO network with ResNet. Based on the Darknet network structure for feature extraction, ResNet is added for feature extraction, which solves the problem of poor accuracy in object detection. Furthermore, in terms of the industrial usage, this proposed network is useful. That is, using solely YOLO model has difficulties in classifying points by their quality, and using image classification model only has difficulties in visualizing where the defect points are located. In this manner, taking only the advantages of both object detection and classification models enable to first detect the heat staking points, and then use classifier to classify whether the detected defect points are defected or not. Please refer to the lines 216 to 226 for changes.
Comment 6: Table 1, caption is not clear, the hyperparameters are no defined, there are more than five!!
Response: Thank you for the comment. We agree that we have not given descriptions and values for the hyperparameters used in the study. Following your suggestion, we have extended the Table 1 and included descriptions as well as values for the hyperparameters. Please refer to the Table 1. We have also included the Table 1 into the response letter.
Comment 7: Table 2, there are no discussion.
Response: Thank you for the comment. We agree that the previously submitted manuscript lacked descriptions of the Table 2. We further added descriptions into the manuscript. Table 2 presents the results. The results show that YOLOv5m outperformed YOLOv5n, YOLOv5s, YOLOv5l, and YOLOv5x, a precision of 0.947, F1-score of 0.930, and mAP@.5 of 0.956. These results are significant. It should be noted that the performance of YOLO v5m model outperforms YOLO v5l and YOLO v5x model. Theoretically, YOLO v5l and YOLO v5x should outperform the YOLO v5m model as v5l and v5x are larger size in terms of the number of extracted features thus are able to train deeper. However, the results show that the optimum number of features to be extracted do not monotonically increase. This im-plies that there are a smaller number of features to be extracted from heat staking points images, and that a certain number of features provided by YOLO v5m model is enough. Therefore, we use the optimal model, YOLO v5m, as the baseline object detection model which is then used to merge with the classification model. Please refer to the lines 264 to 275 for modifications.

Reviewer 3 Report
In this study, we applied an object detection algorithm, the You Only Look Once (YOLO) framework, and a classification algorithm, residual network (ResNet), on a real image 14 heat staking process image dataset. The results look encouraging and motivating. But there are still some contents, which need be revised in order to meet the requirements of publish. A number of concerns listed as follows:
(1) In the introduction, the authors should clearly indicate the contributions and innovations of this paper.
(2) Please highlight your contributions in the Section 1 of introduction.
(3) The computation complexity of the proposed method should be clearly described.
(4) The summary is a little messy, and it is recommended to write it by category.
(5) More statistical methods are recommended to analyze the experimental results.
(6) In the Section 4 of Experimental Results, the values of parameters could be a complicated problem itself, how the authors give the values of parameters? Authors should give these values.
(7) In the Figure 5, the black is same?
(8) The article can be further enhanced by connecting the undergoing work with some existing literatures. For example, 10.1109/JSTARS.2021.3059451; 10.1016/j.ins.2022.08.115; 10.1016/j.ymssp.2022.109422 and 10.3934/mbe.2023090 and so on.
Author Response
We greatly appreciate the review of our paper and are grateful for the referees’ insightful and constructive comments. We have addressed all the comments and below, we outline our point-by-point response to each comment. After the revisions, the paper has improved significantly.
Responses to reviewer 3
Comment 1&2. In the introduction, the authors should clearly indicate the contributions and innovations of this paper. Please highlight your contributions in the Section 1 of introduction.
Response: Thank you for the suggestion. We agree that we have not indicated innovations and contributions on the paper in the introduction section. Therefore, following your suggestion we have described the technical innovations and the contributions of the research in the introduction of the paper.
This paper highlights the technical innovation in the deep learning field. By joining the YOLO and ResNet models, this paper shows a novel method to more accurately detect inspection points and classify their quality simultaneously at an accurate and reliable method.
Our research makes the following contributions: First, our research applies a deep learning framework to a real-time problem, particularly in the heat staking process. Inspecting the quality of a manufactured product and guaranteeing high quality for customers are critical for a business’s sustainable growth. Manual inspection has long encompassed problems. Immature work skills because of frequent labor changes and in-creased process complexity are typical examples that lead to human errors in the inspection process. With the necessity to employ a deep learning-based vision system into the manufacturing process, our paper provides an insight that the combination of YOLO and ResNet can reduce costs and ultimately increase productivity. This paper also contributes to the literature on applying ensemble methods, particularly by combining objection detection methodologies with classification methodologies. This study shows that the model’s performance is improved to the extent that it could be used on a real-time heat staking process. Please refer to the lines 89 to 105 for modifications.
Comment 3. The computation complexity of the proposed method should be clearly described.
Response: Thank you for the comment. We agree that the previously submitted manuscript did not fully and clearly describe the computation complexity of the proposed YOLO-ResNet method. The model is composed of six main parts: a deep fully convolutional network, region proposal network, ROI pooling and fully connected networks, bounding box regressor, classifier for object detection and ResNet classifier. For consistency, this paper employs a deep fully convolutional network. The input image is put into the initial stage and extracted to 256 * n * n feature map, which is the input of region proposal network and ROI pooling layer. In the region proposal network, there are k anchors with different scales and ratios for each point on feature map. There will be n * n * k candidate windows are ranked according to the score, and then 2,000 candidate windows are obtained through non-maximum suppression. Overall, complexity is O(N^2/2), which is consistent with classic YOLO model. This is because the ResNet classifier is attached as a final layer into the current YOLO network. We deeply appreciate your comment, and please refer to the lines 216 to 226 for the update.
Comment 4. The summary is a little messy, and it is recommended to write it by category.
Response: We deeply appreciate your comment. We agree that the previously submitted manuscript’s conclusion was a little messy. We also agree that it contained little information and lacked stories such as contributions and future research guidelines. We thus have edited and included more stories into the manuscript.
This study used YOLO and ResNet models to detect the heat staking points and classify their quality. In the future, it would be recommended to apply other object detection models or classification models and improve the current models. Furthermore, future re-searchers may also consider adding more defect categories for the heat staking process. The current model only considered whether the detected images are defected, nondefected, or unknown. However, there are multiple defect categories in the heat staking process such as overstaking and understaking. More accurate categorization may be beneficial for industries.
This study has a contribution to the manufacturing industry that suffers from high inspection costs. F1-score of the model is well-above 97%. The result implies that if the model is used in the manufacturing inspection process, it would perform like or better than human labor inspection system. Furthermore, if the model can train multiple defect types of heat staking points, then classifying detected heat staking points into different categories by their defect type would also be possible. Such multi-class object detection and classification hybrid model would be an important asset for manufacturing firm’s sustainable development and growth.
Furthermore, this study has a contribution to the growing deep learning application literature. There are recent studies that employ image classification, multi-strategy particle swarm and ant colony hybrid optimization and optimal search mapping on various fields. The underlying mechanism is to use the modern algorithms and under-stand how to employ those algorithms in a certain domain field. In this manner, this study contributes by showing how YOLO and ResNet model could be used in the manufacturing inspection process. Please refer to the lines 319 to 333 for the updated conclusion of the manuscript.
Comment 5. More statistical methods are recommended to analyze the experimental results.
Response: We appreciate your comment and recommendation. We agree that previously submitted manuscript did not include other performance metrics to analyze the experimental results. Following your suggestion, we have added Table 4 and reported YOLO-ResNet model’s performance metrics.
This paper reports statistics such as recall, false negative rate, precision, false negative rate, and F1-score of the model. It should be noted that the overall performance metrics is generally high and significant. Specifically, important metrics for manufacturing firms such as recall, and precision are approximately 98%. Overall results show that the generated YOLO-ResNet model is a significant model that may somewhat replace human labor inspection process. Thank you for the suggestions, and please refer to the Table 4 and the lines 292 to 299 for changes.
Comment 6. In the Section 4 of Experimental Results, the values of parameters could be a complicated problem itself, how the authors give the values of parameters? Authors should give these values.
Response: Thank you for your comment. We agree that the previously submitted manuscript did not provide values of parameters and their descriptions. Therefore, following your suggestion, we have edited Table 1 and further included all hyperparameter values as well as descriptions on each hyperparameter. To crosscheck with other studies using YOLO architecture, we tried to keep the hyperparameters in a default setting used by the most studies. Please refer to the lines 249 to 257 and Table 1 for modifications.
Comment 7. In the Figure 5, the black is same?
Response: Thank you for the comment. As the number of correctly classified defect points (304, 98, and 156) is relatively large compared to wrongly classified defect points (between 0 to 6), it seems that the Python model visualized those wrongly classified classes in a very similar black color.
Comment 8. The article can be further enhanced by connecting the undergoing work with some existing literatures. For example, 10.1109/JSTARS.2021.3059451; 10.1016/j.ins.2022.08.115; 10.1016/j.ymssp.2022.109422 and 10.3934/mbe.2023090 and so on.
Response: Thank you for your suggestion. Following your suggestion, we have carefully read through the abovementioned studies and realized that they would clarify our study. Therefore, we have included them into our submission. Going through the articles, we found that our study also has a contribution to the growing deep learning application literature. There are recent studies that employ image classification, multi-strategy particle swarm and ant colony hybrid optimization and optimal search mapping on various fields. The underlying mechanism is to use the modern algorithms and understand how to employ those algorithms in a certain domain field. In this manner, this study contributes by showing how YOLO and ResNet model could be used in the manufacturing inspection process. Please refer to the lines 327 to 333 and citations 54 to 57 for modifications.

Round 2
Reviewer 2 Report
The manuscript was improved
Reviewer 3 Report
My comments in the first round of review have all been well answered, and I think the paper can be accepted.